# Hydroscapes: A Useful Metric for Distinguishing Iso-/Anisohydric Behavior in Almond Cultivars

**DOI:** 10.3390/plants10061249

**Published:** 2021-06-19

**Authors:** Carolina Álvarez-Maldini, Manuel Acevedo, Manuel Pinto

**Affiliations:** 1Instituto Ciencias Agro-alimentarias, Animales y Ambientales (ICA3), Campus Colchagua, Universidad de O’Higgins, San Fernando 2840440, Chile; mpinto@uoh.cl; 2Instituto Forestal, Centro Tecnológico de la Planta Forestal, San Pedro de la Paz 7770223, Chile; macevedo@infor.cl

**Keywords:** drought stress, leaf water potential, stomatal regulation, gas exchange, isohydric

## Abstract

As a consequence of climate change, water scarcity has increased the use of the iso-/anisohydric concept with the aim of identifying anisohydric or drought-tolerant genotypes. Recently, Meinzer and colleagues developed a metric for discriminating between iso- and anisohydric behavior called the hydroscape, which describes a range in which stomata control leaf water potential (Ψ) with decreasing water availability, and it is linked to several water-regulation and drought-tolerance traits. Thus, our objective was to test the usefulness of the hydroscape in discriminating between iso- and anisohydric *Prunus dulcis* cultivars, a species that is widely cultivated in Mediterranean central Chile due to its ability to withstand water stress. Through a pot desiccation experiment, we determined that the hydroscape was able to discriminate between two contrasting *Prunus* cultivars; the more anisohydric cultivar had a hydroscape 4.5 times greater than that of the other cultivar, and the hydroscape correlated with other metrics of plant water-use strategies, such as the maximum range of daily Ψ variation and the Ψ at stomatal closure. Moreover, the photosynthesis rates were also differently affected between cultivars. The more isohydric cultivar, which had a smaller hydroscape, displayed a steeper photosynthesis reduction at progressively lower midday Ψ. This methodology could be further used to identify drought-tolerant anisohydric *Prunus* cultivars.

## 1. Introduction

Increases in drought events in Mediterranean climates are a widely known consequence of climate change [1,2]. This is a major concern for central Chile, which has been subjected to droughts for almost a decade [3]. Consequently, this has driven the search for new drought-tolerant species and cultivars suited for Mediterranean central Chile, such as *Prunus dulcis* (Mill.) D.A.Webb, which was historically cultivated in Mediterranean regions due to its ability to withstand water stress [4,5,6]. Furthermore, the identification of drought-tolerant cultivars that can sustain production with reduced irrigation could also decrease the demand for hydric resources and increase water-use efficiency.

The search for more drought-tolerant cultivars that can sustain photosynthesis under water-limited conditions has increased the interest in the iso-/anisohydric concept, which can be used to depict the plant–water relation strategy during water stress, the effects on net assimilation [7,8], and the implications for plant growth performance and survival during droughts. Plants that present isohydric behavior are characterized by their conservative water strategy in which they close their stomata, thus maintaining a constant midday minimum leaf water potential (Ψ_min_), while the soil water content and pre-dawn leaf water potential (Ψ_pd_) decline. On the other hand, in plants that present anisohydric behavior, the stomata remain open as the soil water content declines, allowing a decrease in Ψ_min_ [9,10,11]. In almond, strong daily stomatal regulation was observed in response to drought [12]. Thus, the search for anisohydric cultivars, which maintain open stomata and sustain photosynthesis during drought, has gained interest in species such as *Vitis vinifera* L., where genetic variations for this character have been detected [7,13,14,15]. However, in species such as *P. dulcis*, the identification of drought-tolerant genotypes has been restricted to the characterization of physiological changes in response to specific drought treatments or within a narrow range of soil water potentials (Ψ_soil_) [16,17,18,19]. This impairs the proper characterization of the stringency of stomatal control with respect to plant water status and does not allow an accurate determination of an iso-/anisohydric behavior.

Several metrics have been developed to account for iso-/anisohydric behavior. The most commonly used is the slope of the linear regression fitted to the entire Ψ_min_ vs. Ψ_pd_ trajectory (σ, MPa MPa^−1^) during the drying of the soil drying [20]. Here, σ ranges between zero and one to indicate perfectly isohydric plants or perfectly anisohydric plants, respectively. In addition, the difference of Ψ_pd_–Ψ_min_ in a given day (ΔΨ, MPa) indicates an isohydric or anisohydric species with a small or large ΔΨ, respectively [21]. However, the use of these and other metrics has delivered different rankings of species/genotypes along the iso-/anisohydric continuum, as reported by [9] and [11], among others. The shift of species between iso- and anisohydric is caused by different values of Ψ_soil_ that were obtained in each experiment, indicating that stomatal stringency and hydraulic behavior are not solely determined by the genotype, but also by the environment [22].

Recently, Meinzer et al. [10] developed the hydroscape, a new metric for the stringency of stomatal regulation of leaf Ψ during the drying of soil that incorporates the ranges of Ψ_pd_ and Ψ_min_ over which stomata are effective in controlling leaf Ψ as the soil dries, and it defines an area of leaf Ψ over which the plant is able to sustain CO_2_ assimilation. The calculation of the hydroscape requires a wide range of Ψ_soil_, which reduces the ambiguity in iso-/anisohydric behavior due to differences in environmental conditions. In addition, hydroscapes are strongly correlated with various hydraulic traits related to drought tolerance and can be used to effectively separate species according to their drought-response strategies [10,23,24].

Thus, considering the need for new drought-tolerant almond genotypes in central Chile, our objective was to test the usefulness of the hydroscape in discriminating between the levels of stringency of stomatal regulation during drought in two contrasting almond cultivars.

## 2. Results and Discussion

At the beginning of the substrate desiccation (SD) experiments, the gravimetric substrate water content (GSWC) was 93.3 ± 4.5% for the R20 cultivar and 83.6 ± 1.6% for the S/R20 cultivar. The experiments lasted for 17 and 14 days for the R20 and S/20 cultivars, respectively, until Ψ_pd_ = Ψ_min_ was reached. At this point, the GSWC significantly decreased in both cultivars in the SD treatment to 10.2 ± 4.0% and 11.9 ± 2.3% for R20 and S/R20, respectively (Figure 1).

Regarding the first metric of stringency of stomatal control (Table 1), the slope of the linear regression fitted to the relationship of Ψ_pd_ vs. Ψ_min_ (σ), it was observed that σ ranged between 0.38 Mpa Mpa^−1^ in the S/R20 cultivar and 0.81 Mpa Mpa^−1^ in the R20 cultivar, indicating that neither of them exhibited perfect an-/isohydric behavior. According to the σ values, S/R20 was more isohydric than R20 because it had a σ near zero. On the contrary, the hydroscape area was 4.4 times larger in the S/R20 cultivar (4.25 MPa^2^) than in R20 (0.95 MPa^2^) (Table 1, Figure 2). The hydroscape is a metric that represents the stringency of stomatal control of leaf Ψ as the soil dries and the range of water potential over which plants can sustain CO_2_ assimilation [10]. Thus, cultivars with larger hydroscape values, such as S/R20, are expected to be more anisohydric, which, in this case, is contradictory to the values of σ obtained. However, it is not always that a coincidence is found between these metrics. In fact, previous research reported a lack of correlation between the hydroscape and σ [10,24], and σ showed no relation with metrics of stomatal sensitivity in 44 species [9]. Although σ has been commonly used to assess an-/isohydric behavior, it has been proven that it is inconsistent in ranking species/cultivars over the an-/isohydric continuum, depending on the value of Ψ_pd_ over which σ is measured [14,20,25,26].

The water potential at stomatal closure (P_gs90_) was −3.37 and −1.63 MPa in S/R20 and R20, respectively (Table 1). This metric, which is used to define species’ hydraulic safety margins, has been positively correlated with hydraulic traits, such as P_50_ (Ψ inducing 50% loss of conductivity), P_12_ (Ψ inducing 12% loss of conductivity), and water potential at the turgor loss point (Ψ_TLP_), indicating higher thresholds for xylem cavitation at lower P_gs90_ values [27,28,29]. Accordingly, more negative values of P_gs90_ in S/R20 could indicate that the stomata remained open for longer, thus avoiding xylem embolism and allowing carbon assimilation. However, a direct correlation between P_gs90_ and the hydraulic traits in *P. dulcis* needs further research. In agreement with the results for the hydroscape and P_gs90_, S/R20 showed significantly higher ΔΨ than R20 (*p* < 0.0001) (Table 1), which is consistent with an anisohydric behavior [21]. In addition, irrigated S/R20 plants showed significantly higher values of photosynthetic performance A_irr_ (*p* = 0.0031) and gs_irr_ (*p* = 0.0237) than R20 (Table 1). A positive correlation between anisohydric behavior and photosynthetic performance was previously demonstrated [23,29], which is also in relation to higher V_cmax_ and J values in the irrigated S/R20 plants compared to the R20 plants (Table 2).

The imposition of SD negatively affected the photosynthetic parameters in both cultivars, which was previously reported for *Prunus* spp. cultivars [16,30,31]. Plants on the SD treatment, independently of the cultivar, had significantly decreased V_cmax_ and J compared to the WW plants (Table 2). A significant interaction between the cultivars and treatment was observed for g_m_ (*p* = 0.0128). At the end of the experiment, g_m_ decreased by 83.3% in the S/R20 plants subjected to SD, while no significant differences were observed in R20 between the WW and SD treatments (Table 2). A strong decrease in g_m_ in S/R20 could be related to the higher SLA exhibited in this cultivar (206.0 ± 7.3 cm^2^ g^−1^ in S/R20 vs. 132.1 ± 4.6 cm^2^ g^−1^ in R20), which could indicate more packaged mesophyll cells inducing greater leaf internal resistance to CO_2_ diffusion during drought [32]. This could also explain the stronger reduction in A_N_ in S/R20 (78.6%) than in R20 (69.2%) for SD-treated plants compared to WW plants (see Appendix A for changes in photosynthesis-related parameters during the SD experiments in both cultivars).

Different an-/isohydric behaviors were also observed in the dynamics of the A_N_ decrease during the progression of SD. Thus, the cultivars displayed different behaviors for A_N_ with decreasing Ψ_min_ (Figure 3). The S/R20 cultivar sustained a higher A_N_ rate for longer during the progression of SD (more negative Ψ_min_), while a faster decrease in the A_N_ rate was observed in R20 at higher Ψ_min_ values. In *V. vinifera*, Tombesi et al. [33] also described a steeper reduction in A_N_ and gs at higher Ψ_min_ in the isohydric cultivar Montepulciano in contrast to the anisohydric cultivar Sangiovese. Although *V. vinifera* is considered a drought-tolerant species, an-/isohydric behaviors were observed among cultivars [33], indicating that both anisohydric and isohydric cultivars can display drought tolerance, but different dynamics of stomatal regulation during drought are involved in this drought tolerance. This was also observed now among the *Prunus* cultivars, where the S/R20 cultivar, which had the great hydroscape, can be considered a more drought-tolerant cultivar than R20. Our results agree with the findings of previous research that indicated that anisohydric cultivars can sustain photosynthesis for longer during drought events [20,21,23,24], with the implications of greater hydroscapes, as described by Meinzer et al. [10], and with the results for P_gs90_.

Consequently, the metrics for quantifying the an-/isohydry of the hydroscape area, P_gs90_ and ΔΨ, indicated that S/R20 had an anisohydric behavior compared to R20. This is also in agreement with the dynamics of A_N_ with decreasing Ψ_min_, which showed that S/R20 was able to sustain carbon assimilation at lower Ψ_min_ values during drought. Our results coincide with research reporting a positive correlation between the hydroscape area and hydraulic traits related to drought tolerance [10,24], indicating that the hydroscape is a suitable and consistent metric for assessing plant water regulation strategies in *P. dulcis* during droughts.

Current research regarding the response of *Prunus* cultivars to drought is focused on plant responses to a constrained range of leaf and soil Ψ, which cannot be interpreted accurately because iso- and anisohydric species operate at different values of leaf Ψ. Our results confirm that the hydroscape area is a useful metric for assessing an-/isohydric behavior in *P. dulcis* by characterizing the plant water-use strategies and identifying limits for carbon assimilation. This methodology could be used to identify anisohydric *Prunus* cultivars with higher drought tolerance and lower irrigation requirements, thus increasing the water-use efficiency in fruit production. Future research should, however, include a wider range of *Prunus* cultivars and assess the effects of rootstock/scion interactions on an-/isohydric behavior.

## 3. Materials and Methods

### 3.1. Plant Material and Growth Conditions

The experiments were conducted between November of 2019 and January of 2020 at the experimental nursery in the Institute of Agri-Food, Animal, and Environmental Sciences, University of O´Higgins, Chile (latitude: −34.61°, longitude: −70.99°, elevation: 352 m). The plant material used corresponded to one-year old plants of rootstock ROOTPAC^®^20 (*P. besseyi* Bailey × *P. cerasifera* Ehrh) (hereafter R20) and plants of the variety “Soleta” grafted onto R20 (hereafter S/R20). The plants were donated by Agromillora Sur (Río Claro, Maule, Chile, latitude: −35.19°, longitude: −71.25°) in November of 2019 and cultivated for three months in a shaded greenhouse. The plants were transplanted into 3 L plastic pots (14 cm width × 14 cm length × 18 cm height) using a mixture of peat and perlite in a 1:1 proportion as a substrate. The pots were weighed, and a homogeneous weight was assured between them. The plants were irrigated daily and became acclimated to the greenhouse conditions for one month before the beginning of the drought treatments.

### 3.2. Drought Treatment Application

In January of 2020, at the beginning of the drought treatment application, all pots were irrigated to saturation and covered on top with a plastic film, allowing only the plant stem to come out, to ensure that the water losses corresponded to plant transpiration only. The pot’s weights were recorded daily at midday. The plants of the almond cultivars (R20 and S/R20) were divided into two groups (20 plants each): one well-watered or control treatment (WW) and another non-watered group that was therefore submitted to substrate desiccation treatment (SD). Plants on the SD treatment were irrigated at full container capacity only at the beginning of the experiment. Then, irrigation was interrupted, allowing a progressive decrease in substrate water content. The gravimetric substrate water content (GSWC, %) was calculated according to the following equation:GSWC = [(Pot − Pot_dry_)/(Pot_wet_ − Pot_dry_)] × 100(1)
where Pot is the weight of the pot at each measurement time, Pot_dry_ is the weight of the pot with the substrate dried in an oven until achieving a constant weight, and Pot_wet_ is the weight of the pots at container capacity [34].

### 3.3. Midday and Pre-Dawn Water Potential

During the course of the experiment, the soil water potential was monitored regularly using Ψ_pd_ as a proxy. For the leaf water potential measurements, we randomly selected five plants (each plant was a replicate) from each cultivar (2) and irrigation treatment (20 plants in total). One fully developed leaf was excised from the upper third, and the leaf water potential was immediately measured using a Scholander pressure chamber (PMS Instruments, Albany, NY, USA) [35]. The midday minimum leaf water potential (Ψ_min_) was measured between 13:00 and 14:00 h local time, while Ψ_pd_ was measured between 06:00 and 07:00 h local time. Both Ψ_pd_ and Ψ_min_ were measured on at least seven occasions throughout the experiment until Ψ_pd_ = Ψ_min_ was reached, which was indicative of the limits of the stomatal control of the plant water status.

### 3.4. Metric for Characterizing An-/Isohydry

Three metrics for defining the cultivars’ water-use strategies were calculated. First, the slope of the relationship between Ψ_pd_ and Ψ_min_ (σ) was calculated by using linear regression according to [20].

The second metric corresponds to the maximum range of daily leaf water potential variation (ΔΨ). It was calculated as the maximum difference between Ψ_pd_ and Ψ_min_ during the course of the experiment [21].

For the third metric, the hydroscape area (hereafter, hydroscape), we followed the methodology described by [10]. Briefly, the hydroscape is the region comprising the Ψ_pd_ versus Ψ_min_ regression and a 1:1 line, which is calculated as:Hydroscape = (a × b)/2(2)
where a is the intercept of the Ψ_pd_ vs. Ψ_min_ regression, which represents the most negative Ψ_min_ when Ψ_pd_ = 0, and b is the intersection of Ψ_pd_ vs. Ψ_min_ and the 1:1 line, which is the water potential at Ψ_pd_ = Ψ_min_. Data beyond the point where Ψ_pd_ = Ψ_min_ were removed, corresponding to data beyond where the stomatal closure ameliorates Ψ_min_ with further declines in Ψ_pd_.

The leaf water potential at stomatal closure (P_gs90_) was calculated according to [29]. In brief, the stomatal conductance (gs, described in Section 3.5) was plotted against Ψ_min_ and fitted with a weighted polynomial regression to calculate P_gs90_ using the *fitplc* package in R [36].

### 3.5. Gas Exchange Measurements

Measurements of gas exchange were performed at midday on the same days as measurements of Ψ_pd_ and Ψ_min_ with a CIRAS−3 portable photosynthesis system equipped with a chlorophyll fluorescence module (CFM−3; PP Systems, Amesbury, MA, USA). We selected six fully developed leaves from each cultivar (2) and irrigation treatment (2) (24 plants total). The CO_2_ concentration in the leaf cuvette was adjusted to 400 ppm, the leaf temperature was maintained at 25 ± 1 °C, and the photon flux density (PPFD) was set to 1500 µmol m^−2^s^−1^. The leaves were acclimated to the cuvette conditions for at least 10 min; then, the net photosynthesis (A_N_), stomatal conductance (gs), intercellular CO_2_ concentration (C_i_), and transpiration rate (E) were obtained. The intrinsic water-use efficiency (iWUE) was calculated as the ratio between A_N_ and E. The maximum photosynthetic and stomatal conductance rates (A_irr_ and gs_irr_, respectively) were considered as the A_N_ and gs measured in the plants before the imposition of SD at the beginning of the experiment.

At the end of the SD experiment, we created A_N_–C_i_ response curves for the six plants from both cultivars (2) and irrigation treatments (2) (24 plants total) by using combined gas exchange and chlorophyll fluorescence measurements. Before starting the A_N_–C_i_ curves, each plant was adapted to the leaf cuvette conditions for at least 20 min until the photosynthesis and stomatal conductance values were stable. The first step was measured at a reference CO_2_ concentration (C_a_) of 400 ppm, followed by 300, 200, 100, 50, 150, 250, 350, 600, 900, 1200, and 1500 ppm under saturating light conditions of 1500 µmol m^−2^s^−1^. At each CO_2_ concentration, the chlorophyll fluorescence parameters were recorded. The quantum efficiency of PSII (Φ_PSII_) was calculated as:Φ_PSII_ = (F_m_′ − F_s_)/F_s_(3)
where F_m_′ is the maximal fluorescence induced by a saturating pulse of light (8000 µmol m^−2^s^−1^), and F_s_ is the steady-state fluorescence under light. The A_N_–C_i_ curves were transformed into A_N_–C_c_ curves to estimate the maximum velocity of Rubisco carboxylation (V_cmax_) and the electron transport rate (J) according to the method in [37], assuming that the carboxylation rate was either Rubisco limited (A_c_) or ribulose−1,5-biphosphate (RuBP) (A_j_) limited, as described by the models of Farquhar et al. [38]. The mesophyll conductance (g_m_) was also estimated by using Equation (4) according to [37]:g_m_ = [A × (τ × I_inc_ × Φ_PSII_ − 4 × (A + R_d_)]/[τ × I_inc_ × Φ_PSII_ × (C_i_ − Γ × ) − 4 × (C_i_ + 2 Γ*) × (A + R_d_)](4)
where A, C_i_, the photosynthetically active photon flux density incident on the leaf (I_inc_), and Φ_PSII_ are the input data. Then, τ is the product of α (fraction of incoming light absorbed by the photosystems) and β (partitioning fraction of photons between PSI and PSII) and was estimated by using the model described in [37]. The day respiration rate (R_d_) was taken as 1.5% of V_cmax_ [39]. K_C_, K_O_, and Γ* were calculated using the leaf temperature, as described in [40].

### 3.6. Biomass and Leaf Area Measurements

At the end of the desiccation experiment, three plants per cultivar (2) and irrigation treatment (2) (12 plants in total) were randomly selected for biomass and leaf area measurements. The specific leaf area (SLA) was obtained according to [34].

### 3.7. Experimental Design and Statistical Analysis

The design of the experiment was established with a completely randomized factorial design using the cultivar and irrigation treatment as factors, each with two levels. An experimental unit was constituted by a plant with 20 replicates.

We used linear regression to explore the relationship between Ψ_pd_ and Ψ_min_ by using PROC REG procedure (SAS Institute Inc., Cary, NC, USA).

Differences between cultivars for the ΔΨ, A_irr_, and gs_irr_ traits were assessed using Student’s *t*-test with the Stats package of the R software (version 4.0.2, R Development Core Team 2020). To evaluate the effect of the cultivar and irrigation treatment on the photosynthetic parameters, an ANOVA analysis was performed according to our experimental design by using a PROC GLM procedure (SAS Institute Inc.). Differences among means were determined by using a Tukey (HSD) test for multiple comparisons.

Finally, to assess the relationship between A_N_ and Ψ_min_ during SD, an exponential model for R20 and a mechanistic growth model with three parameters for S/R20 were adjusted by using the PROC NLIN procedure (SAS Institute Inc.) with the Gauss-Newton method through a derivative-free algorithm. All visualizations were made using SigmaPlot 10 (Systat Software Inc., San Jose, CA, USA).

## Figures and Tables

**Figure 1 plants-10-01249-f001:**
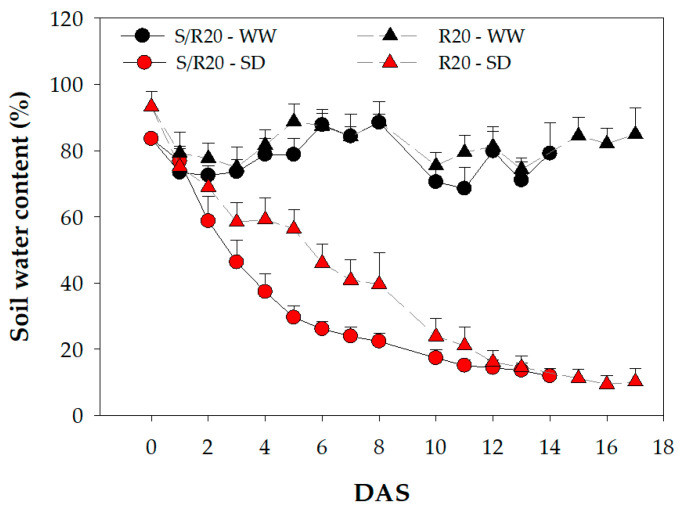
Evolution of the substrate water content (%) during the desiccation experiments for the R20 (triangles) and S/R20 (circles) *Prunus dulcis* cultivars with well-watered treatment (WW, symbols in black) and substrate desiccation treatment (SD, symbols in red). The symbols indicate the mean + standard deviation. DAS: days after stress.

**Figure 2 plants-10-01249-f002:**
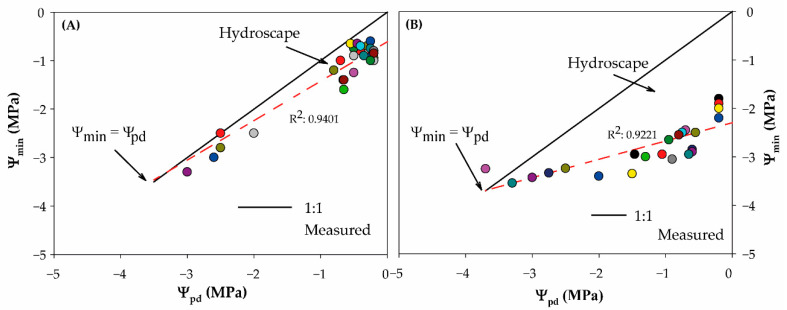
Relationship between pre-dawn water potential (Ψ_pd_) and midday water potential (Ψ_min_) for the R20 (**A**) and S/R20 (**B**) *Prunus dulcis* cultivars. Continuous black lines represent a 1:1 line when Ψ_pd_ = Ψ_min_, and the dashed line represents a fitted linear regression for Ψ_pd_ vs. Ψ_min_. Symbols in colors represent measured values of individual plants during the SD experiments.

**Figure 3 plants-10-01249-f003:**
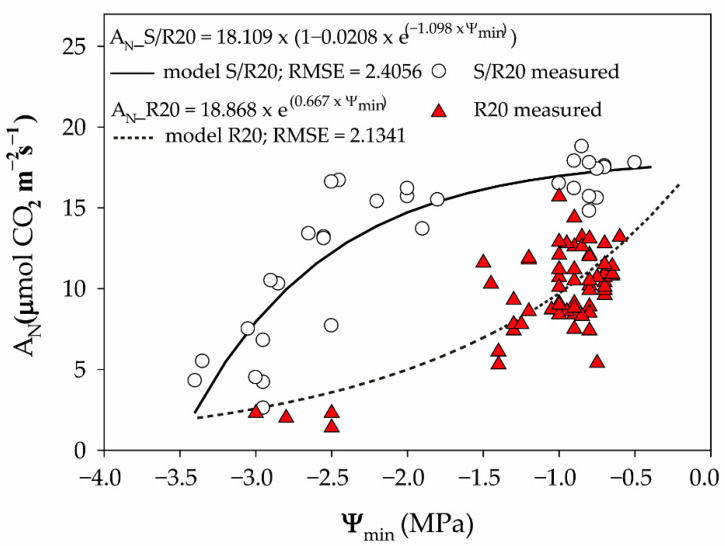
Effect of midday minimum water potential (Ψ_min_) on net photosynthesis (A_N_) during substrate desiccation of the S/R20 and R20 *Prunus* cultivars. Dots and triangles indicate data measured for the S/R20 and R20 cultivars and lines indicate data modeled for S/R20 in a continuous line and R20 in a dashed line. Adjusted models for both cultivars are shown in the panel.

**Table 1 plants-10-01249-t001:** Values of key metrics for describing an-/isohydric behavior in the S/R20 and R20 cultivars. The traits are shown either in mean ± standard error of the mean or values with lower and upper bounds of a 95% confidence interval (bracketed). σ: slope of the relationship between Ψ_pd_ and Ψ_min_; P_gs90_: leaf water potential causing 90% stomatal closure; ΔΨ: maximum range of daily leaf water potential variation; A_irr_: maximal photosynthetic rate in irrigated plants; gs_irr_: maximal stomatal conductance rate in irrigated plants. Different letters indicate significant differences between cultivars at *p* < 0.05 according to Student’s *t*-test.

Trait	Cultivars	
	S/R20	R20
σ (Mpa MPa^−1^)	0.38	0.81
Hydroscape (Mpa^2^)	4.25	0.95
P_gs90_ (−Mpa)	3.37 [3.14; 3.74]	1.63 [1.46; 2.56]
ΔΨ (−MPa)	2.19 ± 0.07 a	1.54 ± 0.04 b
A_irr_ (µmol m^−2^s^−1^)	14.51 ± 0.97 a	10.80 ± 0.16 b
gs_irr_ (mmol m^−2^s^−1^)	389.51 ± 39.41 a	283.52 ± 12.70 b

**Table 2 plants-10-01249-t002:** Mean (± standard error) values of leaf photosynthetic traits measured at the end of the SD experiments, sources of variation, and *p*-values of *Prunus* cultivars subjected to substrate desiccation. V_cmax_: maximum rate of Rubisco carboxylation, J: electron transport rate, g_m_: mesophyll conductance to CO_2_. Different letters indicate statistical differences among means (*p* ≤ 0.05).

	V_cmax_ (µmol CO_2_ m^−2^s^−1^)	J (µmol m^−2^s^−1^)	g_m_ (mmol m^−2^s^−1^)	
			WW	SD
Cultivar (C)				
S/R20	149.57 ± 33.46 a	106.22 ± 13.54 a	208.6 ± 0.094 a	34.8 ± 0.074 b
R20	74.40 ± 3.53 b	81.75 ± 5.79 b	77.3 ± 0.023 b	39.1 ± 0.015 b
Treatment (T)				
WW	108.36 ± 9.44 a	112.96 ± 5.24 a	-	-
SD	72.44 ± 8.88 b	68.83 ± 5.91 b	-	-
Source of variation	*p*-values			
C	0.0034	0.0054	0.0187	
T	0.0014	<0.0001	0.0004	
C × T	0.0708	0.1322	0.0128	

## Data Availability

The datasets generated for this study are available on request to the corresponding author.

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
