# Peer review of "Hydroscapes: A Useful Metric for Distinguishing Iso-/Anisohydric Behavior in Almond Cultivars"

_plants, 2021, doi:10.3390/plants10061249_

Round 1

Reviewer 1 Report

An interesting communication.  The comments below are minor.

Abstract

suggest replace 'This methodology could be further used to search for drought tolerant anisohydric...' with 'This methodology could be further used to identify drought tolerant anisohydric...'

Page 3 please check the following text 'was almost four five larger in the S/R20 cultivar'

Author Response

Response to reviewer 1 comments. 

Point 1: suggest replace 'This methodology could be further used to search for drought tolerant anisohydric...' with 'This methodology could be further used to identify drought tolerant anisohydric...'

Response 1: In the sentence “This methodology could be further used to search for drought tolerant anisohydric…”the works “search” has been replaced by “identify”.

Point 2: Page 3 please check the following text 'was almost four five larger in the S/R20 cultivar

Response 2: In page 3 (line 92), the sentence “was almost four five larger in the S/R20 cultivar” was replaced by “was 4.4 times larger in the S/R20 cultivar”

Reviewer 2 Report

The authors presented interesting studies which showed that hydroscape is a useful metric to discriminate iso/anisohydric behawior in almont cultivars. Drought stress leads to plant growth damage and a decrease in yields. An increase in drought events in Mediterranean climates is a consequence of climate change. There is a need to search for new drought-tolerant species and cultivars suited for Mediterranean central Chile such as Prunus dulcis (Mill.) D.A.Webb. The identification of drought-tolerant cultivars that can sustain production with reduced irrigation is very important for agriculture.

  • The Abstract is well written and reflects the objectives and results obtained. I have noticed a typing error – single bracket.
  • The Introduction part is well written.  Please add information about stomatal diurnal regulation in almond. What is the correct a constant midday leaf water potential (Ѱmin ) or a constant midday minimum leaf water potential (Ѱmin )?

I have two questions:

  • Are anisohydric cultivars drought tolerant in any case? Is there any example of isohydric cultivar which are drought tolerant? This should be discussed.
  • At the end of Introduction section has been written that “two contrasting almond cultivars” were tested, but I have not found in Results or Materials and Methods section information which cultivar is drought tolerant S/R20 or R20?

Author Response

Response to Reviewer 2 Comments

Point 1:  The Abstract is well written and reflects the objectives and results obtained. I have noticed a typing error – single bracket. 

Response 1: The single bracket in line 19 has been deleted.

Point 2: The Introduction part is well written.  Please add information about stomatal diurnal regulation in almond. What is the correct a constant midday leaf water potential (Ѱmin) or a constant midday minimum leaf water potential (Ѱmin)?

Response 2: We specified that Ѱmin corresponds to midday minimum leaf water potential at line 41-42 in the introduction. This was also specified in the materials and methods section at line 220. Also, in lines 44 to 46, a sentence has been included indicating that daily stomatal regulation has been observed in almond in response to drought stress.

Point 3: Are anisohydric cultivars drought tolerant in any case? Is there any example of isohydric cultivar which are drought tolerant? This should be discussed.

Response 3: In the present report, the drought tolerant concept is referred to the ability to sustain higher photosynthesis rates and lower leaf water potential as substrate water content decreased. Particularly, this behaviour was observed in the S/R20 cultivar. This does not mean, that isohydric cultivars or species can´t tolerate or survive water-limited conditions. For example, Vitis vinifera is considered a drought tolerant species, however the range of an/isohydric behaviours have been observed among cultivars of this species. Thus, Cabernet Sauvignon and Shiraz are considered isohydric and anisohydric, respectively. This indicates, that isohydric cultivars can be drought tolerant, however, the stomatal regulation mechanism to survive under water-limited conditions are different from the anisohydric cultivar.

Thus, in lines 37-38 there is a description of what is considered here as drought tolerance. Also, the discussion regarding drought tolerance of isohydric cultivars was added in lines 159-163.

Point 4: At the end of Introduction section has been written that “two contrasting almond cultivars” were tested, but I have not found in Results or Materials and Methods section information which cultivar is drought tolerant S/R20 or R20?

Response 4: This is now specified between lines 162 to 164.  

Reviewer 3 Report

This paper deals with the establishment of the discrimination by the hydroscape in Almond cultivars. The main research findings of this paper will be important for the full evaluation for drought-tolerance cultivars. This paper seems to have been competently conducted.

I have a minor point.

In Table 2, "a1" is modified into "a" ?

Author Response

Response to Reviewer 3 Comments

Point 1: In Table 2, "a1" is modified into "a" ?

Response 1: In table 2 “a1” has been replaced with “a”